# Effects of Fertilization Approaches on Plant Development and Fertilizer Use of Citrus

**DOI:** 10.3390/plants11192547

**Published:** 2022-09-28

**Authors:** Xiaochuan Ma, Feifei Li, Yuewen Chen, Yuanyuan Chang, Xuefei Lian, Yunsong Li, Li Ye, Tao Yin, Xiaopeng Lu

**Affiliations:** 1College of Horticulture, Hunan Agricultural University, Changsha 410128, China; 2National Center for Citrus Improvement, Changsha 410128, China; 3Institute of Horticulture, Hunan Academy of Agricultural Science, Changsha 410125, China

**Keywords:** citrus, fertigation, plant development, fruit quality, fertilizer use efficiency (FUE), fertilizer loss

## Abstract

Fertilization is an important part of citrus crop management. However, limited details are available about the fertilization approach on citrus plant development. A pot experiment for the fertilization approaches and fertigation levels were conducted in this study. Four fertilization approaches, namely, drip fertigation (DF), broadcast fertilization (CK^+^), hole fertilization (HF) and pour fertilization (PF) were tested. The fertigation level treatment included 100% (DF-337.5), 80% (DF-270), 60% (DF-202.5) and 40% (DF-135) fertilizer supply with DF, and the 100% fertilizer supply with broadcast fertilization were served as control (CK). The results showed that DF not only increased the absorptions of nitrogen (N), phosphorus (P) and potassium (K) but also promoted citrus plant height, stem diameter and dry weight. In fruit quality, DF had the highest fruit total soluble solid (TSS) and titratable acidity (TA) contents. For fertilizer loss, DF had the lowest N and K leaching losses of 9.26% and 4.05%, respectively, and the lowest N and K runoff losses among the approaches. Isotopic tracing with ^15^N indicated that DF had the highest fertilizer use efficiency. Based on the analysis of fertigation levels, DF approach with 60% fertilizer reduction could improve citrus plant development. Therefore, DF promoted citrus plant growth and fruit quality by accelerating fertilizer utilization and impairing fertilizer loss. The fertilizer amount in citrus production could be reduced significantly using DF.

## 1. Introduction

Citrus is a major global fruit crop [1,2]. In 2019, China had a total citrus yield of 45.84 million tons and plantation area of 2.61 million ha (http://data.stats.gov.cn/index.htm, accessed on 5 August 2022). Water and fertilizer are crucial for citrus growth and fruit yield. The amount of fertilizer in citrus management is associated with production cost and environmental problems. Most citrus-producing areas around the world recommend fertilizer amounts of 200–350 kg/ha N, 100–200 kg/ha P and 150–450 kg/ha K. In Florida, the optimum N amount in orange is 260 kg/ha [3]. As determined through years of trials in Iran, the appropriate amounts of N, P and K for orange are 150–250, 100–150 and 150–200 kg/ha, respectively [4]. In China, a survey of fertilizer application through the main citrus-producing provinces revealed that the annual amounts of N, P and K in citrus are 485, 198 and 254 kg/ha, respectively [5]. Excessive fertilizer application results in fertilizer wastes and environmental problems. N and P fertilizers running off with rainwater pollute surface water and cause water eutrophication. In Tulare County in the U.S., statistical data indicates that elevated nitrate level in domestic well water is strongly associated with the management of citrus orchards [6]. The agricultural discharges of N and P cause water eutrophication, thus disrupting the normal supply of drinking water in shore cities [7]. Additionally, salt ions in fertilizer could be trapped in the soil, and then the excess salt ions cause soil salinization and poison crops [8]. In the past decades, broadcast, hole and pour fertilization approaches were used mainly in most Chinese citrus orchards, and a series of problems were exposed also. In combination with the increment of labor cost, drip fertigation instead of conventional approaches is becoming mainstream. 

Drip fertigation is an effective technology for both irrigation and fertilization. It was first developed and applied in Israel in 1960s, and then rapidly spread to the other places of the world. Drip fertigation technology results in the infiltration of soluble fertilizer into the crop root area and maintains an appropriate water and fertilizer supplies [9]. In comparison with conventional fertilization approaches, drip fertigation has the advantages in promoting the use efficiencies of water and fertilizer and improving crop yield and quality [9,10]. Drip fertigation increases the leaf N and stem diameter of peach trees with a N fertilizer saving for 25% to 50% [11]. Drip fertigation of potato fields reduces NO_3_^−^-N leaching [12]. In the citrus orchards of Florida, drip fertigation reached 7% of fertilizer savings but 9% and 8% increases in yield and fruit total soluble solid (TSS) [13]. Drip fertigation on tomatoes could increase P and K transport and reduce NO_3_^−^-N leaching from soil [14]. Drip fertigation can effectively improve crop yield and quality. In comparison with the conventional approaches, drip fertigation increases the yield of apple by 29.2%, titratable acidity (TA) by 4.9% and TSS by 5.3% [15]. It also promotes date palm yield by 41% and saves 66% of fertilizers [16].

Citrus in China is mainly planted in the hills, where water and fertilizer retain poorly. With the increment of labor force cost, drip fertigation in citrus industry is becoming more popular, but its specific affection has not been fully quantified. In the present study, the effects of four fertilization approaches, including drip fertigation, on fertilizer use efficiency, citrus plant growth and fruit quality were studied in detail. The results of this study provide a guide for suitable fertilization and irrigation in China’s citrus industry.

This study evaluated the advantages of DF in citrus production comprehensively. The data quantified that DF was economical and effective to plant development and fruit quality formation relative to the conventional fertilization approaches. DF not only impaired the leaching and runoff losses of fertilizers but also improved FUE significantly. In DF approach, approximate 60% fertilizer reduction was possible in citrus production.

## 2. Results

### 2.1. Effects of Fertilization Approaches on Citrus Plant Growth and Fruit Quality

Fertigation improved the vegetative growth of citrus plant. drip fertigation (DF) and pour fertilization (PF) approaches had significantly higher stem diameter than the other fertilization approaches (Figure 1A). However, plant height showed no significant difference among the approaches (Figure 1B). Among the treatments, DF had the highest plant dry weight, in which stem and fruit dry weights were significantly higher than those in other approaches. Among the approaches, hole fertilization (HF) and without fertilization (CK^−^) had the lowest dry matters of root, fruit and total plant dry weight (Figure 1C).

In plant nutrient, plants with DF had the most NPK accumulations. Among the fertilization approaches, DF made the highest NPK contents in stem and fruit, while broadcast fertilization (CK^+^) had the highest NPK contents in leaf. In each fertilization approach, N accumulated most in the root, while enrichment in fruit was the lowest. K accumulated mainly in fruit and leaf, but only 30% was observed in the stem and root. All plant tissues had low P content. Overall, the N:P:K ratio in the whole citrus plant was close to 1:0.06:0.14 (Table 1). DF promoted medium- and micro-nutrients absorptions. For Ca, Zn and B, the contents in DF approach reached two-fold in other fertilization approaches. Plant Mg and Mn in DF treatment were also remarkably higher than those in other fertilization approaches. However, these medium- and micro-nutrients showed no difference among CK^+^, HF and PF approaches. Some nutrients were even lower than plants with CK^−^ (Figure 1D).

Relative to conventional fertilization approaches, DF remarkably improved fruit quality. Totally, fruits in water-soluble fertilization approaches (DF and PF) exposed higher TSS than non-water-soluble approaches (CK^+^ and HF). Fruits in DF showed the highest TSS and TA. Except for CK^−^, fruits with DF had higher VC than those in other fertilization approaches (Figure 2). Among the approaches, fruits in HF had the lowest TSS and VC.

### 2.2. Effects of Fertilization Approaches on NPK Leaching and Runoff Losses

DF approach had 9.26% leaching loss of N and 4.05% leaching loss of K, while leaching loss in HF reached 15.17%. Unlike NK leaching, soil P had the minimum leaching loss in CK^+^ approach and the maximum leaching loss in PF approach. Although leaching loss differed across the fertilization approaches, the approximate leaching loss rates of NPK were 11.16%, 0.04%, and 4.61%, respectively. Relative to leaching loss, less runoff loss was observed in all fertilization approaches. Among the approaches, HF exhibited the lowest runoff loss in NPK. Approximately, the runoff loss rates of NPK were 0.29%, 0.01%, and 0.05% (Table 2). Overall, soil N had the highest loss, but P had the lowest loss in either leaching or runoff losses.

### 2.3. Soil N Mobility under Different Fertilization Approaches

Through ^15^N isotopic tracing, a soil depth of 30 cm was crucial for N absorption, and DF approaches promoted N enrichment at this depth. At the fruit enlargement stage, CK^+^ approach resulted in ^15^N enrichment at a soil depth of 30 to 50 cm. CK^+^ and DF approaches had significantly higher ^15^N content at a soil depth of 30 cm than that in 10 and 50 cm (Figure 3A). In all approaches, ^15^N was concentrated mainly in 30 cm depth soil. At the soil depth of 50 cm, DF approach exhibited the lowest ^15^N content, while CK^+^ approach enriched most. HF approach made ^15^N retained most at a soil depth of 10 cm. At the fruit maturity stage, soil ^15^N declined to 50% of that at development stage in all layers. The content of ^15^N increased overall along with soil deepening. At the soil depth of 10 and 30 cm, HF approach had the highest ^15^N content, but it remarkably declined at the maturity stage. At the soil depth of 50 cm, no^15^N difference was observed among the approaches (Figure 3B).

### 2.4. Nitrogen Use Efficiency in Citrus Plant with Different Fertilization Approaches

Ndff% reflects the transportation and utilization of N in plant. Among the approaches, Ndff% increased progressively from roots to stems and then to leaves. The fruit exhibited the highest Ndff%. Ndff% also indicated the use efficiency of N fertilizer. Among the approaches, DF had the highest use efficiency of ^15^N, reaching 13.61%. The ^15^N use efficiency of DF was mainly contributed by the fruit, and the value was remarkably higher than that of other approaches. PF made the highest ^15^N content in plant stems and leaves (Table 3). 

### 2.5. Effects of Reducing Fertilizer on Citrus Plant Development

Chemical fertilizer application can be reducible for citrus plant growth. Among the treatments, DF-337.5 and CK^+^ had the lowest plant dry weight, but the highest plant dry weight was observed in DF-135 treatment. Shoot numbers had no difference between treatments in spring, summer and autumns, but the diameter differed depending on treatment. Stem diameter and dry weight increased with the decrease in fertilizer amount in each season. In autumn, DF-135 treatment had the maximum shoot diameter (Table 4). Within a range, NPK absorptions increased with the decrease in fertilizer amount. DF-135 had the highest plant NPK content and relative increment. By contrast, DF-337.5 had the lowest plant NPK contents (Figure 4). 

## 3. Discussion

Citrus is generally planted in south of China, which is characterized by rolling terrain. With the changes in industrial environment, citrus production in China is facing problems, including fertilization difficulty, increasing fertilizer and labor cost, and higher fertilizer loss. Accordingly, fertigation has become increasingly popular, and it can be an effective solution. In the present work, fertigation and conventional fertilization approaches were compared in terms of fertilizer utilization and citrus plant development.

### 3.1. Citrus Growth, Mineral Nutrition and Fruit Quality

Water and fertilizer are crucial for plant growth, development, yield, and fruit quality. A good supply of water and fertilizer will make plant vigorous and fruiting well [17]. Different combinations of fertilization and irrigation affect citrus growth [18]. In comparison with furrow irrigation, drip fertigation increases the stem volume and biological yield of poplars [19]. Drip fertigation results in higher maize production than conventional fertilization [20]. Similar results were found in the present study, in which DF approach significantly increased citrus plant stem diameter and plant dry weight (Figure 1A,C). The advantage in nutrient supplies by DF may play an important role in plant development [21,22]. In comparison with manual fertilization and surface irrigation, drip fertigation in tomato increases the N and K content by 45.6% and 45.3%, respectively [23]. In comparison with chemical fertilizers to soil, water and fertilizer coupling promotes soil fertility, in which drip fertigation is much better than pour fertilization [22,24]. Sufficient nutrition from DF accelerates plant growth in terms of the trunk diameter and dry matter [25]. In the present work, DF fertilization helped the plant accumulate more NPK in the root, stem, fruit or total plant (Table 1).DF also promoted the absorptions of medium- and micro-nutrients in citrus plant (Figure 1D). Therefore, DF promoted citrus plant and fruit development in terms of the increased efficiency of nutrient supply. In combination with the soil condition of rolling terrain in south of China, DF is really a good option for citrus crops. In addition, the N:P:K ratio in citrus plants was close to 1:0.06:0.14 in all fertilization approaches, indicating a specific NPK pattern in citrus plants. The ratio may provide a guideline for NPK supplies to citrus plants.

Fruit quality is an important concern for citrus fruit, especially for fresh citrus fruit consumption in many countries. Fertilization approaches are important options for fruit quality improvement [26,27]. In comparison with conventional fertilization approaches, fertigation increased the fruit sugar content by 7.3% in apple [28]. Fertigation also improved fruit TSS and juice percent of citrus [29]. In the present study, fruit TSS, TA and VC contents with DF increased by 13.4%, 17.6% and 27.5%, respectively, compared with CK^+^ (Figure 2). With the increases of fruit quality components, NPK accumulated more in fruit (Table 1). The results indicated that N promoted the synthesis and accumulation of sugars through the intensity of photosynthesis, while P and K were involved fruit sugar or sugar-acid ratio [30]. Therefore, DF approach improves citrus fruit quality through the enhancement of plant NPK absorptions.

### 3.2. Fertilizer Loss 

Fertilizer spreads via leaching and runoff cause pollution of groundwater, rivers, lakes and other water sources [31,32]. Fertigation could reduce the leaching of soil NO^3−^-N [12], reduce leaching loss of nitrate nitrogen, and improve FUE of citrus [18] and tomato [14]. In fertigation, the fertilizer mainly penetrates the soil layers at the depth of 0 to 30 cm [9], where the citrus roots are distributed mainly. Leaching and runoff fertilizer losses are the main fertilizer loss ways in citrus crop production, and fertigation minimizes fertilizer loss. In the present study, DF approach had the lowest leaching loss, implying higher fertilizer use (Table 2).

### 3.3. Nitrogen Use Efficiency 

N is important for tree development, fruit yield and quality [3,33]. During plant growth, N in stems and leaves is gradually transferred to storage organs such as cotyledons and fruits, resulting in the highest N in matured fruit. By contrast, plant root absorbs the least N because of transportations to the aboveground tissues [34]. Fertigation helps N movement and use in fruit crops. With drip fertigation, nitrogen use efficiency (NUE) increases by 24.7% in citrus [35]. In comparison with furrow fertilization, drip fertigation increases NUE by 94.7% in tomato [10]. As N pool of plant, fruit enriches a large amount of N through plant absorption, and the increment of N in fruits promotes NUE in drip fertigation [35]. In isotope tracing assay, DF resulted in the highest fruit Ndff% in all fertilization approaches, thus promoting NUE in citrus directly (Table 3). In soil, N was remarkably enriched at a soil depth of 30 cm in all fertilization approaches, where citrus roots grow mainly (Figure 3A), and this part could be used well during plant development (Figure 3B). These results indicated that DF improved N enrichment at the soil depth of 30 cm and N transfer efficiency from soil to plant, in which fruit N accumulation contribute greatly. 

### 3.4. Fertilizer Amount

By using drip fertigation, tomato yield increased after 25% reduction of fertilizer supply [14]. In comparison with conventional fertilization, fertigation in citrus could reduce fertilizer by 25%, and it will increase FUE by 119% and fruit yield by 64% [29]. In eggplant, drip fertigation reduced fertilization by 20% without change of plant growth, but the fruit yield and plant FUE were improved well [36]. In the present study, DF-135 treatment with 60% fertilizer reduction exposed the highest stem diameter, dry weight, plant NPK contents (Table 4) and the highest plant NPK absorptions (Figure 4). These results indicated that the decrease in fertilizer amount was available in improvements of citrus plant growth and FUE by using fertigation approach.

## 4. Materials and Methods

### 4.1. Material and Plant Growth Conditions

In fertilization approach treatment, four-year-old ‘Succari’ sweet orange (*Citrus.sinennsis* L. Osbeck) trees grafting on trifoliate orange (*Poncirus trifoliata* L. Raf.) rootstock were used at Changsha (28° 10′ 47″ N, 113° 4′ 37″ E), Hunan Province, China. The actual mean maximum temperature ranged from 28°C to 35°C, and the mean minimum temperature in growing months varied from 5°C to 12°C. Stainless steel barrels with height of 60 cm and diameter of 40 cm were ordered specifically for this investigation. Due to the roots of citrus plant enrich in 20 to 40 cm soil depth generally, so the barrel depth was sufficient for plant growth. A leaching outlet with diameter of 5 cm was designed at the bottom of the barrel and a runoff outlet with diameter of 3 cm was designed at the soil surface of the barrel. Liquid from bottom and top outlets was collected as leaching and runoff water, respectively. The planting medium in the barrel was composed of 70% yellow clay, 15% organic matter and 15% sawdust. Urea labeled with ^15^N was obtained from Shanghai Research Institute of Chemical Industry (Shanghai, China).

Three-year-old ‘Juxiangzao’ Satsuma mandarin (*Citrus unshiu* Marc.) trees were used in fertilizer amount reduction experiment. Citrus tree was planted in a plastic pot with a top diameter of 35 cm, bottom diameter of 27 cm and height of 27 cm. Water-soluble fertilizer (Stanley, China) containing 20% N, 20% P, 20% K, 0.1% Fe, 0.05% Zn, 0.05% Mn, 0.01% Mo and 0.05% B was used in all experiments.

### 4.2. Experimental Design for Fertilization Approach Experiment and Isotopic Experiment

Four fertilization approaches including CK^+^, DF, HF and PF were studied, and the set-up without fertilization was named CK^−^. This experiment was conducted in open air with rainfall. In CK^+^, fertilizers were spread evenly on the soil surface in planting bucket. DF was performed in a fertigation system (insert the dripping arrow into the soil) with water-soluble fertilizer supply. For HF, three holes with depth of 15 cm were made in an equilateral triangle, and fertilizers were buried into the holes. In PF, fertilizers were dissolved in water and poured evenly into the soil. The recommended dose of fertilizers (75:75:75 g NPK/tree/year) were supplied in each approach treatment, except for CK^−^. Through a year, 30%, 20% and 50% of fertilizers were applied at the shooting, fruit set and fruit enlargement stage, respectively. At the plant shooting stage, fertilizers were supplied in early March and mid-April. At the fruit set stage, fertilizers were applied in mid-May and mid-June. At the fruit enlargement stage, fertilizers were supplied in early July, early August and early September. Leaching water and runoff water were collected, and all mineral nutrients were measured throughout the year. After the experiment, all plants were excavated and divided into root, stem, leave, and fruit for dry weight, nutrient and fruit quality measurements. At fruit maturity, fruits were collected from the periphery of the canopy, and five fruits were selected for fruit quality analysis. Five replicates were employed for each approach, and one tree was selected for each replicate.

Isotope was used to test nutrient movement from soil to plant here. A half gram ^15^N urea per plant was added in each fertilization approach described above. Soil samples were collected at 10, 30 and 50 cm depths in middle July and early December. Each barrel was sampled with two holes, and then the samples were dried, crushed and sieved through 80 mesh. In December, all plants were excavated and divided into tissues for ^15^N measurement. The abundances of ^15^N in soil and plant were determined using isotope mass spectrometry [37]. Ndff (%) = (^15^N atom% in plant sample − ^15^N atom% in the nature)/(^15^N atom% in fertilizer − ^15^N atom% in the nature) × 100; ^15^N atom% in the original medium is 0.366% in this study. ^15^N use efficiency (%) = [Ndff% × total nitrogen in organs (g)]/fertilization amount (g) × 100.

### 4.3. Experimental Designs for Fertilizer Reducing Experiment

In the fertilizer reduction experiment, 337.5 g water-soluble fertilizer (DF-337.5, containing 67.5: 67.5: 67.5 g NPK) was supplied initially to ‘Juxiangzao’ over a year depending on citrus plant requirement [38]. Then, 20% fertilizer application reduction was set as a gradient, including 80% fertilizer supply (DF-270), 60% fertilizer supply (DF-202.5) and 40% fertilizer supply (DF-135). The control was involved in broadcast fertilization of 337.5 g fertilizer around a year. This experiment was conducted in a greenhouse, and all plants were irrigated every 3 to 7 days depending on temperature and rainfall. Three replicates were employed for each approach, in which one tree was selected for each replicate.

### 4.4. Determinations of Mineral Nutrition and Fruit Quality

Alkaline N was determined using the alkaline hydrolysis diffusion absorption method [39]. Available P was extracted using sodium bicarbonate and measured by Molybdenum-Antimony anti-spectrophotometry method [40]. Available K was extracted with ammonium acetate [39], and it was measured by inductively coupled plasma mass spectrometry (ICP-MS) [41]. 

Plant samples were washed with deionized water and inactivated by heating to 105 °C for 15 min. The inactivated samples were dried at 65°C in a forced-air oven to constant weight, ground manually, and stored for further analysis. Plant samples were digested using hydrogen peroxide and sulfuric acid for NPK determinations [42]. N was determined using the Kjeldahl method [43]. P was extracted and determined spectrophotometrically as blue molybdate-phosphate complexes [44], and K was measured via flame photometry. NPK increment = NPK contents of plants after test − NPK contents of plants before test. For Ca, Mg, Mn, Cu, Zn and B assays, plant samples were digested using dry ash method [45] and determined via ICP-MS [41]. 

Total N in water was determined by alkaline potassium persulfate digestion-UV spectrophotometry [46]. Total P in water was determined by sulfuric acid-hydrogen peroxide elimination-vanadium-molybdenum yellow colorimetric method [46]. Total K in water was determined by sulfuric acid-hydrogen peroxide elimination-flame photometric method [47]. NPK leaching/runoff loss efficiency (%) = (total NPK leaching or runoff loss in each treatment − total leaching or runoff loss in CK^−^)/NPK application amount.

Fruit quality, including TSS, TA and vitamin C (VC) were immediately assayed after sampling. Fruit TSS was measured using the juice with a refractometer (PAL-BX, ATAGO). Fruit TA was determined by titration with an aliquot of juice against 0.1 M NaOH [48]. VC content was determined using iodometric method [49]. 

### 4.5. Statistical Analysis

Values are presented as mean ± SD. Data were subjected to analysis of variance (ANOVA) in IBM SPSS Statistics 23.0. Different letters indicate significant differences at *p* < 0.05 (Duncan’s test).

## 5. Conclusions

In comparison with conventional fertilization, DF promoted the absorptions of N, P, K, Ca, Mg, Zn and B in citrus plant. Through the increased efficiency of nutrient supply, plant development and fruit quality were improved in DF approach. Furthermore, the N:P:K ratio in the whole citrus plant was close to 1:0.06:0.14. DF promoted the enrichment of N fertilizer at the soil depth of 30 cm and increased the efficiency of nitrogen utilization. The approximate leaching loss rates of NPK were 11.16%, 0.04% and 4.61% respectively, while the runoff loss rates of NPK were 0.29%, 0.01% and 0.05% respectively. Relative to conventional fertilization, 60% fertilizer amount reduction with DF approach was available to improve citrus plant growth and FUE. Basing on what we obtained in this study, the more economical and effective fertilization strategies for citrus crop would be made in the future. Subsequently, we are more interested in knowing how drip fertigation running over years affects the plant development and the physicochemical properties of orchard soil.

## Figures and Tables

**Figure 1 plants-11-02547-f001:**
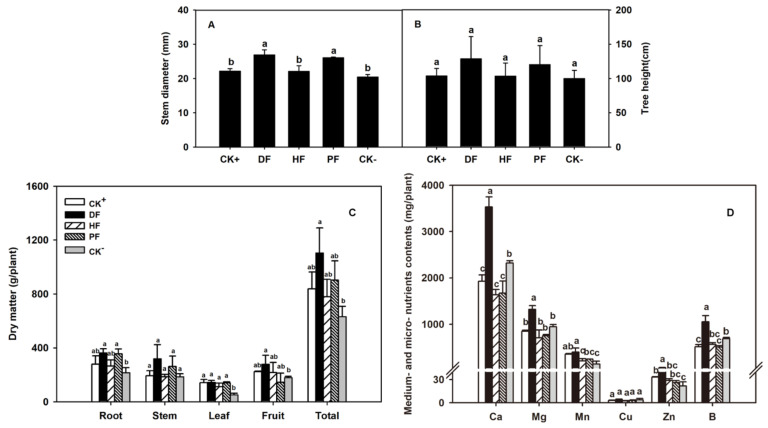
Growth of citrus plants under different fertilization approaches: (**A**) Stem diameter, (**B**) tree height, (**C**) dry matter and (**D**) medium- and micro- nutrients. The data are presented as mean ± SD, *n* = 3.Different letters indicate significant differences at *p* < 0.05, Duncan’s test.

**Figure 2 plants-11-02547-f002:**
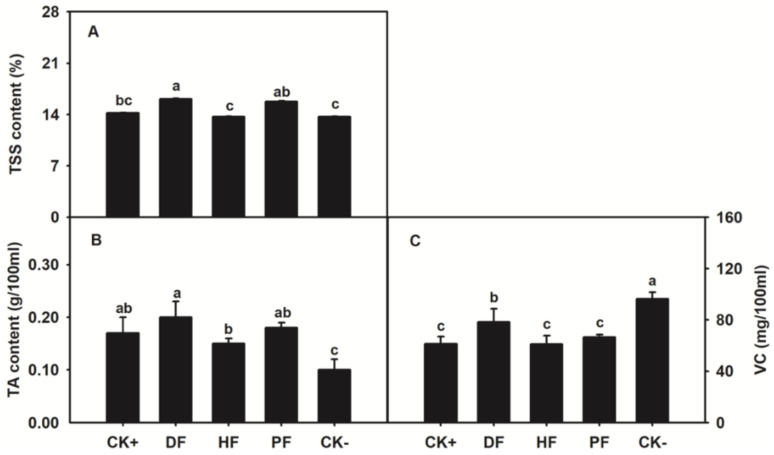
Effects of fertilization approaches on citrus fruit quality. (**A**) TSS, (**B**) TA and (**C**) VC contents. The data are presented as mean ± SD, *n* = 3. Different letters indicate significant differences at *p* < 0.05, Duncan’s test.

**Figure 3 plants-11-02547-f003:**
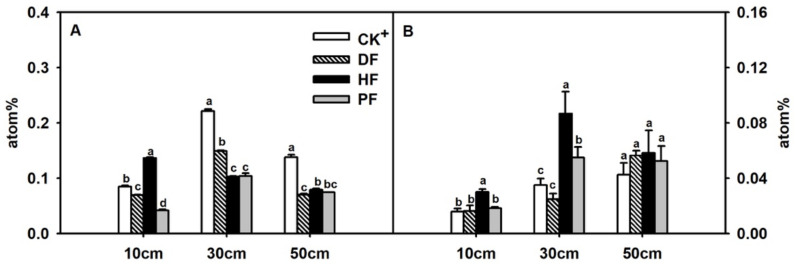
Enrichment of ^15^N in different soil depth at (**A**) citrus fruit enlargement and (**B**) fruit maturity stages. The data are presented as mean ± SD *n* = 3. Different letters indicate significant differences at *p* < 0.05, Duncan’s test.

**Figure 4 plants-11-02547-f004:**
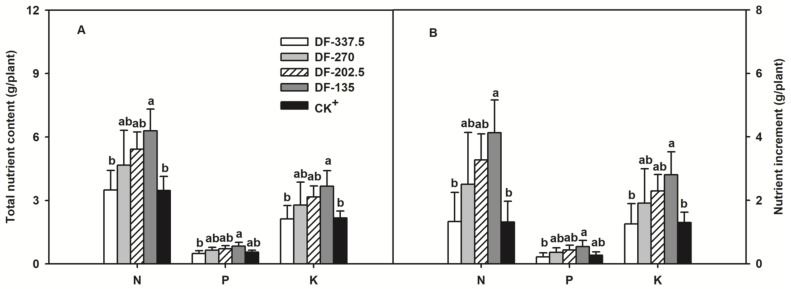
NPK absorptions in citrus plant under fertigation levels. (A) Total nutrient content and (**B**) nutrient increment in different fertigation levels. The data are presented as mean ± SD *n* = 3. Different letters between treatments indicate significant differences at *p* < 0.05. NPK increment = NPK contents of plants after test − NPK contents of plants before test.

**Table 1 plants-11-02547-t001:** NPK contents in plant tissues under different fertilization approaches.

	Treatment	Root (g/Plant)	Stem (g/Plant)	Leaf (g/Plant)	Fruit (g/Plant)	Total (g/Plant)
N	CK^+^	9.10 ± 1.11 ^a^	3.49 ± 0.37 ^b^	5.97 ± 0.59 ^a^	3.75 ± 0.29 ^ab^	20.02 ± 4.13 ^b^
	DF	10.22 ± 0.55 ^a^	6.26 ± 0.23 ^a^	5.12 ± 0.60 ^a^	4.43 ± 0.55 ^a^	27.18 ± 2.29 ^a^
	HF	7.09 ± 0.53 ^b^	3.28 ± 0.26 ^b^	3.83 ± 0.21 ^b^	2.98 ± 0.16 ^b^	18.18 ± 1.12 ^b^
	PF	10.08 ± 0.60 ^a^	4.33 ± 1.52 ^b^	5.29 ± 0.35 ^a^	3.85 ± 0.93 ^ab^	20.21 ± 0.47 ^b^
	CK^−^	1.94 ± 0.18 ^c^	0.98 ± 0.06 ^c^	0.92 ± 0.18 ^c^	1.46 ± 0.40 ^c^	6.67 ± 0.17 ^c^
P	CK^+^	0.44 ± 0.12 ^a^	0.24 ± 0.02 ^bc^	0.28 ± 0.02 ^a^	0.38 ± 0.04 ^b^	1.47 ± 0.02 ^ab^
	DF	0.44 ± 0.07 ^a^	0.47 ± 0.03 ^a^	0.23 ± 0.02 ^b^	0.53 ± 0.07 ^a^	1.66 ± 0.19 ^a^
	HF	0.46 ± 0.07 ^a^	0.22 ± 0.003 ^c^	0.18 ± 0.02 ^c^	0.29 ± 0.07 ^bc^	1.16 ± 0.01 ^b^
	PF	0.47 ± 0.05 ^a^	0.36 ± 0.10 ^ab^	0.25 ± 0.02 ^ab^	0.30 ± 0.07 ^bc^	1.18 ± 0.06 ^b^
	CK^−^	0.22 ± 0.06 ^b^	0.16 ± 0.03 ^c^	0.08 ± 0.01 ^d^	0.24 ± 0.03 ^c^	0.70 ± 0.14 ^c^
K	CK^+^	0.81 ± 0.21 ^b^	0.88 ± 0.18 ^ab^	4.16 ± 0.26 ^a^	3.05 ± 0.09 ^b^	8.53 ± 1.80 ^ab^
	DF	1.68 ± 0.25 ^a^	1.83 ± 0.70 ^a^	2.40 ± 0.30 ^bc^	4.20 ± 0.58 ^a^	10.85 ± 2.17 ^a^
	HF	0.78 ± 0.13 ^b^	0.79 ± 0.001 ^b^	1.97 ± 0.47 ^c^	3.16 ± 0.41 ^b^	7.26 ± 0.64 ^b^
	PF	2.03 ± 0.48 ^a^	1.54 ± 0.03 ^ab^	2.90 ± 0.28 ^b^	3.20 ± 0.32 ^b^	9.15 ± 0.01 ^ab^
	CK^−^	0.33 ± 0.09 ^b^	0.77 ± 0.07 ^b^	0.73 ± 0.06 ^d^	1.76 ± 0.35 ^c^	3.37 ± 0.79 ^c^

Different letters between treatments of one tissue indicate significant differences at *p* < 0.05.

**Table 2 plants-11-02547-t002:** NPK leaching and runoff losses in different fertilization approaches.

	Treatment	Total Leaching Loss in Yield Cycle (mg)	leaching Loss Efficiency (%) ^(1)^	Total Runoff Loss in Yield Cycle (mg)	Runoff Loss Efficiency (%) ^(1)^
N	CK^+^	94.19 × 10^2^ ± 4.74 × 10^2 b^	10.38 ± 0.63 ^b^	7.26 × 10^2^ ± 0.81 × 10^2 a^	0.28 ± 0.11 ^a^
	DF	85.73 × 10^2^ ± 1.44 × 10^2 b^	9.26 ± 0.19 ^b^	7.03 × 10^2^ ± 0.08 × 10^2 a^	0.25 ± 0.01 ^a^
	HF	130.09 × 10^2^ ± 12.45 × 10^2 a^	15.17 ± 1.66 ^a^	7.06 × 10^2^ ± 0.22 × 10^2 a^	0.26 ± 0.03 ^a^
	PF	89.88 × 10^2^ ± 2.14 × 10^2^ b	9.81 ± 0.29 ^b^	8.26 × 10^2^ ± 0.48 × 10^2 a^	0.42 ± 0.06 ^a^
	CK^−^	16.28 × 10^2^ ± 1.70 × 10^2 c^		5.13 × 10^2^ ± 0.46 × 10^2 b^	
P	CK^+^	20.17 ± 1.92 ^d^	1.80 × 10^−2^ ± 0.30 × 10^−2 c^	9.24 ± 0.45 ^b^	0.90 × 10^−2^ ± 0.10 × 10^−2 b^
	DF	31.00 ± 0.47 ^c^	3.20 × 10^−2^ ± 0.10 × 10^−2 b^	18.69 ± 1.38 ^a^	2.20 × 10^−2^ ± 0.20 × 10^−2 a^
	HF	50.59 ± 4.88 ^b^	5.90 × 10^−2^ ± 0.70 × 10^−2 a^	2.26 ± 0.08 ^c^	-
	PF	58.91 ± 6.31 ^a^	7.00 × 10^−2^ ± 0.80 × 10^−2 a^	18.19 ± 3.00 ^a^	2.10 × 10^−2^ ± 0.40 × 10^−2 a^
	CK^−^	6.51 ± 0.63 ^e^		2.49 ± 0.30 ^c^	
K	CK^+^	33.06 × 10^2^ ± 1.40 × 10^2 c^	4.22 ± 0.19 ^c^	54.48 ± 5.89 ^a^	6.90 × 10^−2^ ± 0.80 × 10^−2 a^
	DF	31.83 × 10^2^ ± 1.84 × 10^2 c^	4.05 ± 0.25 ^c^	31.26 ± 4.51 ^c^	3.80 × 10^−2^ ± 0.60 × 10^−2 b^
	HF	42.60 × 10^2^ ± 2.17 × 10^2 a^	5.49 ± 0.29 ^a^	25.14 ± 4.55 ^c^	3.00 × 10^−2^ ± 0.60 × 10^−2 b^
	PF	36.58 × 10^2^ ± 0.59 × 10^2 b^	4.69 ± 0.08 ^b^	45.81 ± 3.55 ^b^	5.80 × 10^−2^ ± 0.50 × 10^−2 a^
	CK^−^	1.42 × 10^2^ ± 0.14 × 10^2 d^		2.61 ± 0.37 ^d^	

Different letters between treatments of one element indicate significant differences at *p* < 0.05. ^(^^1)^ NPK leaching/runoff loss efficiency (%) = (Total NPK leaching or runoff loss in each treatment − Total leaching or runoff loss in CK^−^)/NPK application amount.

**Table 3 plants-11-02547-t003:** N use efficiency under different fertilization approaches.

	Root Ndff% ^1^	Stem Ndff%	Leaf Ndff%	Fruit Ndff%	^15^ N Content (g/Plant)	^15^ N Use Efficiency (%) ^2^
CK^+^	19.00 × 10^−2^ ± 1.10 × 10^−2 a^	21.20 × 10^−2^ ± 1.50 × 10^−2 ab^	26.00 × 10^−2^ ± 1.00 × 10^−2 ab^	32.80 × 10^−2^ ± 1.60 × 10^−2 b^	5.60 × 10^−2^ ± 1.70 × 10^−2 ab^	11.29 ± 3.40 ^ab^
DF	18.70 × 10^−2^ ± 0.80 × 10^−2 a^	19.80 × 10^−2^ ± 0.80 × 10^−2 b^	24.50 × 10^−2^ ± 0.30 × 10^−2 b^	35.70 × 10^−2^ ± 0.50 × 10^−2 a^	6.80 × 10^−2^ ± 0.20 × 10^−2 a^	13.61 ± 0.43 ^a^
HF	17.40 × 10^−2^ ± 0.50 × 10^−2 a^	21.50 × 10^−2^ ± 0.60 × 10^−2 ab^	22.00 × 10^−2^ ± 0.80 × 10^−2 c^	25.80 × 10^−2^ ± 0.20 × 10^−2 c^	3.80 × 10^−2^ ± 0.30 × 10^−2 b^	7.64 ± 0.61 b
PF	19.40 × 10^−2^ ± 0.10 × 10^−2 a^	23.30 × 10^−2^ ± 0.50 × 10^−2 a^	26.90 × 10^−2^ ± 1.00 × 10^−2 a^	32.40 × 10^−2^ ± 01.70 × 10^−2 b^	5.90 × 10^−2^ ± 0.60 × 10^−2 ab^	11.79 ± 1.23 ^ab^

Different letters between treatments indicate significant differences at *p* < 0.05. ^1^ Ndff (%) = (^15^N atom% in plant sample − ^15^N atom% in the nature)/(^15^N atom% in fertilizer − ^15^N atom% in the nature) × 100; ^15^N atom% in the original medium is 0.366% in this study. ^2 15^N use efficiency (%) = [Ndff% × total nitrogen in organs (g)]/fertilization amount (g) × 100.

**Table 4 plants-11-02547-t004:** Citrus plant growth with reducing fertilization amount using fertigation.

	Plant Height (cm)	Stem Diameter (mm)	Plant Dry Weight (g)	Spring Shoot Number	Summer Shoot Number	Autumn Shoot Number	Spring Shoot Stem Diameter	Summer Shoot Stem Diameter	Autumn Shoot Stem Diameter
DF-337.5	105.23 ± 15.04 ^a^	13.06 ± 1.05 ^b^	177.66 ± 37.00 ^c^	21.60 ± 9.07 ^a^	16.60 ± 5.85 ^a^	18.50 ± 10.52 ^a^	2.64 ± 0.21 ^b^	2.82 ± 0.35 ^ab^	3.22 ± 0.51 ^ab^
DF-270	115.53 ± 4.35 ^a^	13.63 ± 0.62 ^b^	202.45 ± 37.08 ^bc^	20.40 ± 8.33 ^a^	18.80 ± 9.09 ^a^	18.75 ± 8.79 ^a^	3.02 ± 0.34 ^a^	2.84 ± 0.25 ^ab^	3.66 ± 1.40 ^ab^
DF-202.5	117.43 ± 6.01 ^a^	12.74 ± 0.47 ^b^	254.28 ± 36.14 ^b^	29.20 ± 16.38 ^a^	25.00 ± 10.02 ^a^	29.00 ± 7.35 ^a^	2.75 ± 0.25 ^ab^	3.02 ± 0.22 ^a^	3.31 ± 0.25 ^ab^
DF-135	125.60 ± 11.79 ^a^	15.07 ± 0.50 ^a^	377.43 ± 23.79 ^a^	13.80 ± 9.91 ^a^	17.40 ± 4.03 ^a^	20.80 ± 3.97 ^a^	2.71 ± 0.15 ^ab^	2.90 ± 0.29 ^ab^	4.02 ± 0.29 ^a^
CK^+^	107.60 ± 7.93 ^a^	12.95 ± 0.23 ^b^	183.43 ± 21.37 ^c^	24.40 ± 16.62 ^a^	7.40 ± 7.17 ^b^	20.75 ± 13.92 ^a^	2.53 ± 0.15 ^b^	2.59 ± 0.15 ^b^	2.76 ± 0.40 ^b^

Different letters between treatments indicate significant differences at *p* < 0.05.

## Data Availability

Not applicable.

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
