# Peer review of "Effects of Fertilization Approaches on Plant Development and Fertilizer Use of Citrus"

_plants, 2022, doi:10.3390/plants11192547_

Round 1
Reviewer 1 Report
Very interesting article that shows a lot of interesting results. Protection of water and soils from over-fertilization is a very topical issue. The article proves that with a well-composed combination of fertigation and fertilization, it is possible to significantly reduce the dose of fertilizers, as well as increase the use of nutrients.
Article is written in accordance with existing rules for writing articles. Discussion of results correctly done.
I list my comments below, which should be taken into consideration:
General comments:
1. The quality of the English language must be improved. There are many errors in the text that need to be corrected. I suggest proofreading with a qualified translator.
2. Throughout the manuscript, please correct the units: kg/hm2 to kg/ha (kilograms per hectare - a unit used in agronomy).
3. Please add an abbreviation list
Introduction:
Line 46: Eutrophication refers to surface water, not groundwater. The position presented by the authors is not true
Line 57: In Israel not by Israel
Line 77: Please add a statement of novelty
Results:
Line 100: The average was used to represent the study groups. Please demonstrate in the statistical analysis that the results had a normal distribution. If the distribution was not normal, please use the median.
Line 104: Please detail for the reader whether to interpret statistically significant letters/differences vertically or horizontally.
Line 125/Table 2: Please put the significant digits in order. Using even 7 significant digits is not correct. This also applies to other tables.
Line 173/Figure 4 – Please add the description of letters a, b, c….
Materials and methods:
Line 256 and 257: Celcius degrees are written behind their value here (contiguous), with no additional space.
Line 268: Please avoid using micromineral elements, because it is an unclear group of elements.
Line 311: Please describe the determination method of available P, as done for K or N.
Line 333: The description of the statistical analysis is too general and raises many questions. On the other hand, if only ANOVA and Duncan's test were performed then the analysis was performed incorrectly and the results of such an analysis cannot be the basis for publication.
I ask for a comprehensive dissection of the path of statistical analysis, with an indication of the size of the groups (equal/unequal), assessment of the normality distribution, assessment of homogeneity of variance, assessment of statistically significant differences, indicating the p-value.
Line 338: Sentence is unclear. Please rewrite.
Conclusions:
Line 337: Please add some future perspectives to your study.
References:
Please correct the references and use more current items. Almost half of the references are older than 12 years. Among the 49 items, there are also items of 1960, 1975, 1965 or 1978 - this is unacceptable.
Reviewer 2 Report
The results are clearly mentioned in the article, which show DF promoted citrus plant growth and fruit quality by accelerating fertilizer utilization and impairing fertilizer loss.
However, I am concerned about the possibility of different growing conditions in the field and in pot trials.
In particular, there may be more root zone limitations in pots than in the field. I would think that the 60 cm depth of the pot would be sufficient.
If you have measurements, could you indicate the depth of the root zone in the pot trial?
Or you could add a statement such as the pot depth was sufficient for citrus growth. 
Reviewer 3 Report
The current manuscript describes a significative work aimed to understand the fertilization approachs on citrus plant development. The amount of fertilizer in citrus management is associated with production cost and environment problems. Meanwhile, citrus in China is mainly planted in the hills, where water and fertilizer are retained poorly. Hence, the present study aiming to clear the effects of four fertilization approaches, including drip fertigation, on fertilizer use efficiency, citrus plant growth, and fruit quality. The results of this study provide a guide for suitable fertilization and irrigation techniques for citrus planting in China. This is a good and important study for citrus production.
Specific comments:
1. In Figure 1D and Figure 4, the author should clearly explain the significance between CK and treatment. I suggest that the same element in different treatments should be put together, so as to better reflect the comparability.
2. The notes in Tables 1 and 2 should account for significant differences, and it should be written clearly that different letters between different treatments of the same element represent significant differences.
3. I suggest the author should recreate the Figure 2.
Round 2
Reviewer 1 Report
I accept the manuscript in the current version. I am satisfied with the responses to the comments I made. I have no further objections.